# Conduction Conditions for Self-Healing of Metal Interconnect Using Copper Microparticles Dispersed with Silicone Oil

**DOI:** 10.3390/mi14020475

**Published:** 2023-02-18

**Authors:** Naoki Suetsugu, Eiji Iwase

**Affiliations:** School of Fundamental Science and Engineering, Waseda University, 3-4-1 Okubo, Shinjuku-ku, Tokyo 169-8555, Japan

**Keywords:** self-healing metal interconnect, copper microparticles, silicone oil, voltage, current

## Abstract

This study clarifies the conditions for the bridging and conduction of a gap on a metal interconnect using copper microparticles dispersed with silicon oil. An AC voltage applied to a metal interconnect with a gap covered by a dispersion of metal microparticles traps the metal microparticles in the gap owing to the influence of a dielectrophoretic force on the interconnect, thus forming a metal microparticle chain. The current was tuned independently of the applied voltage by changing the external resistance. An AC voltage of 32 kHz was applied to a 10 µm wide gap on a metal interconnect covered with 3 µm diameter copper microparticles dispersed with silicone oil. Consequently, the copper microparticle chains physically bridged the interconnect and exhibited electrical conductivity at an applied voltage of 14 V_rms_ or higher and a post-bridging current of 350 mA_rms_ or lower. It was shown that the copper microparticle chains did not exhibit electrical conductivity at low applied voltages, even if the microparticle chains bridged the gap. A voltage higher than a certain value was required to achieve electrical conductivity, whereas an excessive voltage caused bubble formation and destroyed the bridges. These phenomena were explained based on the applied voltage and reference value of the current after bridging.

## 1. Introduction

Cell/metal/semiconductor microparticles with diameters of a few micrometers or less are trapped between the interconnect upon the application of AC voltage between the interconnect. This phenomenon is referred to as “electric field trapping,” which has been used to separate specific microparticles [1,2], produce micro- and nanostructures of microparticles [3,4,5,6,7], and measure the properties of quantum dots [8,9,10,11,12,13,14]. A self-healing metal interconnect that uses electric field trapping to repair cracks in metal interconnects has been proposed [15,16,17]. The application of voltage to an interconnect covered with a liquid dispersed with metal microparticles results in the generation of an electric field around the disconnected area. Consequently, dielectrophoretic forces act on the microparticles, trapping them in the disconnected area. The number of trapped microparticles increases, and the microparticles eventually form a chain that fills the disconnected area and causes conduction at the interconnect. The metal microparticle dispersion is left untouched after conduction. Previous studies on self-healing metal interconnects have used gold microparticles dispersed with an aqueous solution [15,16,17]. Due to the possible effects of the oxide film on copper microparticles and the complex permittivity of copper microparticles and silicone oil, we investigated the conduction conditions for a self-healing metal interconnect with a gap covered by silicone oil dispersed with copper microparticles. The aim was to clarify the conditions under which the copper microparticle chains bridged the gap of the interconnect.

Direct observations and electrical measurements of a metal interconnect were performed by varying the applied voltage as well as the current generated after bridging. We tuned the current independently of the applied voltage by changing the external resistance. Thereafter, the behaviors of the copper microparticle chains in the bridging and conducting forms were compared through the application of low voltage, high voltage with low current, and high voltage with high current. In addition, the copper microparticle chain formation process was directly observed using a microscope.

## 2. Examination of Bridging and Conduction Morphology

Copper microparticles were subjected to a dielectrophoretic force from a non-uniform electric field generated by the application of AC voltage to an interconnect and were trapped in the disconnected area (Figure 1a). However, when the copper microparticles were covered with an oxide film, the oxide film functioned as an insulating layer, and the copper microparticles bridged the interconnect but were not electrically conductive (Figure 1b). If the oxide film on the copper microparticles can be broken, the copper microparticle chains should exhibit electrical conductivity (Figure 1c). For example, by increasing the applied voltage, the oxide film on copper microparticles can be broken by dielectric breakdown or Joule heating generated between the copper microparticles before conduction occurs. However, in the case of a large current flowing through the copper microparticles through conduction at the interconnect, the degree of Joule heating generated at the copper microparticle bridge will also be excessively high. The high degree of Joule heating would result in the boiling of the silicone oil around the copper microparticle chains, which would destroy the copper microparticle chain, thus preventing conduction (Figure 1d). Therefore, while applying a voltage capable of breaking down the oxide film of the copper microparticles with Joule heating generated before conduction, self-healing is possible only if the silicone oil around the copper microparticle chain does not boil due to Joule heating. The energy of Joule heating can be evaluated by the current flowing after microparticle chain conduction [17]. Therefore, in this study, we fabricated a gold interconnect with a 10 µm wide gap on a glass substrate, as shown in Figure 2a, and evaluated whether the gap could be bridged and/or conducted by changing the values of the applied voltage and current. In this study, 32 kHz was selected as the frequency of the applied AC voltage based on a preliminary test. In the case of an aqueous solution dispersed with gold microparticles, it has been reported that the growth rate of metal particle chains is fast, ranging from 10 kHz to 100 kHz, owing to AC electro-osmotic flow and other effects [3]. Although the dispersoid and dispersion media were different in the present study, we examined the velocity of the copper microparticles in the range of several tens of kHz and chose 32 kHz as the frequency of the applied AC voltage. The dimensions of the gold interconnect and the material parameters of silicone oil and the copper microparticles are listed in Table 1, Table 2 and Table 3. A 0.01 µm thick chromium layer was used as the adhesion layer, and a 0.5 µm thick gold layer was deposited on a glass substrate. The deposited gold was patterned via photolithography in the shape of a 25 µm wide interconnect with a gap of 10 µm. Silicone oil dispersed with 3 µm diameter copper microparticles was used to cover the gap in the interconnect. Furthermore, an external resistance was connected in series with the interconnect, and AC voltage was applied to both the interconnect and external resistance using a waveform generator and voltage amplifier for approximately 5 min. In addition, by varying both the resistance *R*_res_ of the external resistance and the voltage *V*_all_ applied to the circuit, the voltage *V*_crack_ applied to the interconnect and the current flowing through the trapped copper microparticles were varied. Thus, the level of Joule heating generated after self-healing was varied. The *R*_res_ values were 4600, 1600, 1000, 460, 300, and 100 Ω. When a microparticle chain bridges the crack, *V*_crack_ decreases to *V*_crack_ because the impedance of the cracked gold interconnect decreases significantly. The value of *V*_all_/*R*_res_ was used as the representative value of the current flowing in the microparticle chain. The microparticles were observed under a microscope (VW-600C, Keyence) to determine whether they had bridged the interconnect. The impedance |*Z*| of the interconnect was measured by connecting an LCR meter (ZM2376, NF Corporation) to the interconnect after voltage was applied, as shown in Figure 2b. Since the absolute value of the impedance dropped rapidly when the copper microparticles were conducted through the interconnect, the degree of conduction was determined based on the impedance value.

Figure 3 shows the classification of the conditions under which the copper microparticles bridged and conducted an interconnect. By changing the external resistance, only the current was changed from the state where the same voltage was applied. In addition, the plots with constant external resistance are connected by dotted lines. From the plots shown in Figure 3a, we classified three types of bridging and conduction patterns observed when copper-microparticle-dispersed silicone oil was used for the self-healing of the metal interconnect, and clarified that a certain voltage must be applied to self-healing while the current flowing after conduction must be controlled below a certain level. Type 1 was defined as a case in which the applied voltage was 21 V_rms_ or lower. Type 2 was defined as the case wherein the applied voltage was 14 V_rms_ or higher and the reference value of the current was less than 350 mA_rms_. Finally, Type 3 was defined as a case in which the reference value of the current was 350 mA_rms_ or higher. Figure 3b shows the change in the impedance of the interconnect when the external resistance was maintained at a constant value of 100 Ω and the applied voltage was varied. As shown in Figure 3b, when the applied voltage was 7.1 V_rms_ (Type 1), the impedance was 10^6^ Ω, indicating that no electrical conduction was established. When the applied voltage was between 14 and 28 V_rms_ (Type 2), the impedance was 10^1^ Ω, indicating that electrical conduction had been established. Finally, when the applied voltage was between 35 and 57 V_rms_ (Type 3), the impedance was 10^6^ Ω, indicating that electrical conduction was not achieved. Thus, when the external resistance was set to 100 Ω, electrical conduction was not achieved for Types 1 and 3, that is, applied voltages of 7.1 V_rms_ and 35–57 V_rms_, respectively. However, electrical conduction was obtained only when the applied voltage was in the range of 14–28 V_rms_ (Type 2) and self-healing was achieved. Figure 3b,c show the impedance and photographs of Type 1, respectively. The results regarding Type 1 show that electrical conduction was not achieved (as can be seen from Figure 3b); however, the microparticles bridged the interconnect (as can be seen from Figure 3b). Figure 3b,d show the impedance and a photograph of Type 2. The results regarding Type 2 show that electrical conduction was achieved (as can be seen from Figure 3b), and microscopic observation showed that the copper microparticles bridged the interconnect (as can be seen from Figure 3d). Thus, Type 2 is a self-healing type, wherein copper microparticles bridge the interconnect and facilitate conduction. The oxide film of the copper microparticles in Type 1 was an insulating layer that was bridged but not conductive. Moreover, the oxide film of the copper microparticles was destroyed by increasing the voltage, resulting in electrical conduction, as in Type 2. Finally, Figure 3b,e show the impedance values and photographs of Type 3. In Type 3, electrical conduction was not established (as can be seen from Figure 3b). Further, a microscopic observation (Figure 3e) showed that bubbles were generated from the bridging area of the interconnect and that the copper microparticles did not bridge the interconnect afterwards. In addition, the interconnect was destroyed by bubbles. Thus, based on these results, it was clarified that when copper microparticle-dispersed silicone oil was used for the self-healing of the metal interconnect, the bridging and conduction of the interconnect could be explained based on the applied voltage and reference value of the current flowing after the bridging.

## 3. Direct Observation of Microparticle Chain Behavior

To clarify the differences in the states of the copper microparticle chains of Types 1, 2, and 3, the process of copper microparticle chain formation was observed using a microscope and subsequently compared. A gold interconnect with a gap of 10 µm in width was fabricated on a glass substrate. A drop of silicone oil containing copper microparticles with a diameter of 3 µm was placed over the interconnect, and an AC voltage was applied with varying external resistance. An oscilloscope was connected in parallel to the interconnect to monitor the voltage across the interconnect. If the copper microparticles renders the interconnect conductive, the voltage *V*_crack_ applied to the interconnect also decreases because of the decrease in impedance. The decrease in the voltage from *V*_crack_ to *V*_crack_’ observed by the oscilloscope was considered to indicate self-healing. First, an external resistor of 100 Ω was connected, such that the reference value of the current was 71 mA_rms_ at an applied voltage of 7.1 V_rms_ (Type 1), and the interconnect was observed and animated using a microscope. Next, an external resistance of 4600 Ω was connected to the interconnect to allow 9.2 mA_rms_ of current (referent value) to flow at an applied voltage of 42 V_rms_ (Type 2), and the interconnect was observed and animated using a microscope. Finally, an external resistor of 100 Ω was connected such that the reference value of the current was 566 mA_rms_ at an applied voltage of 57 V_rms_ (Type 3), and the interconnect was observed and animated using a microscope. A microscope was used to photograph the interconnect from the top, and a metal halide lamp was used to illuminate the microscope from the top and bottom. The frame rate of the microscope was 200 fps, the shutter speed was 1/1000 s, and magnification was 3000×.

Figure 4, Figure 5 and Figure 6 show a series of optical images of the copper microparticle chain’s behavior at low voltages taken to observe Type 1; those taken at high-voltage and a low reference value of current to observe Type 2; and those taken at a high-voltage and a high reference value of current to observe Type 3. Figure 4a shows the disconnected area before the application of 7.1 V_rms_. The copper microparticles had already precipitated following the dispersion of a drop of silicone oil with copper microparticles. When an external resistance of 100 Ω was set and a low voltage of 7.1 V_rms_ was applied, the copper microparticles bridged the interconnect at 153.455 s, as shown in Figure 4b. However, after 300 s, when the copper microparticles bridged the interconnect but did not achieve electrical conduction, they gathered but did not self-heal. Figure 5a shows the disconnected area before the application of 42 V_rms_. In this state, when the external resistance was set to 4600 Ω and a high voltage of 42 V_rms_ was applied, the copper microparticles bridged the interconnect at 127.320 s, as shown in Figure 5b. In this experiment, the electric field generated between the interconnect disappeared because of electrical conduction, as shown in Figure 5b, and the microparticles were not attracted beyond Figure 5b. Figure 6a shows the disconnection area before the application of 57 V_rms_. When a high voltage of 57 V_rms_ was applied with an external resistance of 100 Ω, the copper microparticles bridged the interconnect at 22.190 s, as shown in Figure 6b. However, a bubble was generated around the bridging area (22.195 s), as shown in Figure 6c. Subsequently, the bubble spontaneously disappeared, and the bridged copper microparticle chains around the location where the bubble was generated were destroyed. In addition, the surrounding interconnect to which the bridging area was connected was destroyed and chipped. Bubbles were generated every time the copper microparticles bridged the interconnect, thus destroying the bridges and causing the bubbles to disappear. As for the mechanism of bubble generation in the case of high voltage with a high reference value of current, Joule heating owing to the current led to the generation of the copper microparticles that bridged the interconnect and rapidly heated the area around the bridges, and the energy of Joule heating per unit time was larger than the energy released from the microparticles to the medium. Consequently, silicone oil was heated locally and boiled. Subsequently, the current temporarily stopped flowing and no bubbles were generated.

## 4. Conclusions

This study clarified the bridging and conduction conditions that occurred when copper microparticle-dispersed silicone oil was used for the self-healing of a metal interconnect. By changing the applied voltage and current, which was assumed to be flowing after conduction (adjusted by changing the external resistance), bridging and conduction were classified into three types. Below a certain voltage, the oxide film of the copper microparticles bridged the gap on the interconnect but did not conduct (Type 1). Notably, bridging and conduction require a certain voltage (Type 2). However, if the current flowing after conduction exceeds a certain level, bubbles are generated from the bridging area of the copper microparticles and are then destroyed (Type 3). The bridging and conduction morphologies of each type were examined through microscopic observations of the copper microparticle chains to clarify their behavior when low voltage, high voltage with low current, and high voltage with high current were applied. The copper microparticles were trapped and bridged on the interconnect when a low voltage and high voltage with low current were applied. When a high-voltage with low current was applied, the copper microparticles were conducted through the interconnect, but no bubbles were observed. When high-voltage with high current was applied, bubbles were generated, and the copper microparticles bridged the interconnect and broke the copper microparticle chains.

## Figures and Tables

**Figure 1 micromachines-14-00475-f001:**
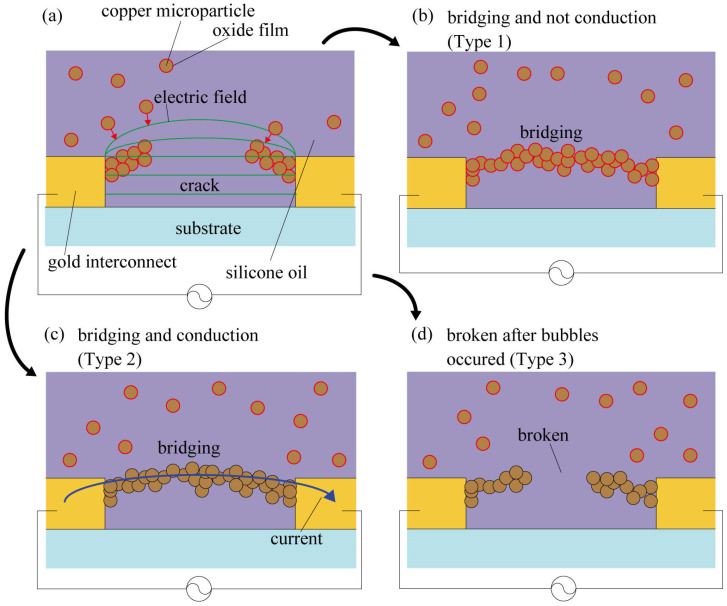
Schematic image of self-healing using copper microparticles dispersed with silicone oil. (**a**) Electric field trapping of copper microparticles in crack of gold interconnect via dielectrophoresis. (**b**) State that achieves physical bridging and does not achieve electrical conduction by an oxide film of copper microparticles (Type 1). (**c**) State that achieves both physical bridging and electrical conduction (Type 2). (**d**) State wherein copper microparticle chain is broken by high voltage and high current (Type 3).

**Figure 2 micromachines-14-00475-f002:**
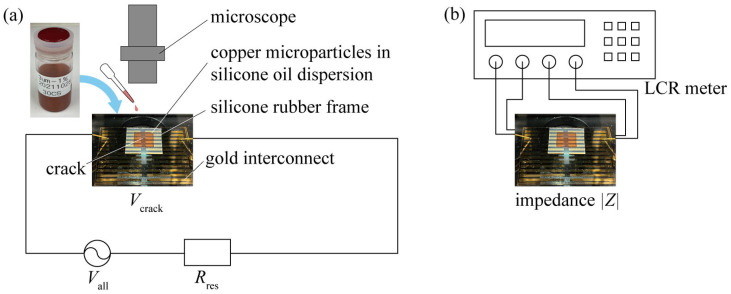
Experimental setup for evaluation of bridging and conduction morphology of copper microparticles. (**a**) Observations regarding physical bridging. (**b**) Impedance measurement for electrical conduction.

**Figure 3 micromachines-14-00475-f003:**
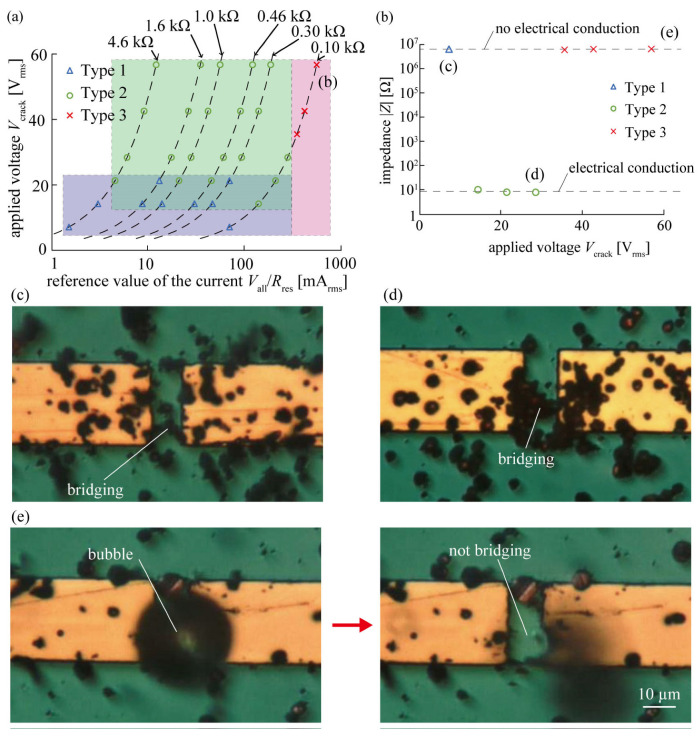
Classification of bridging and conduction morphology of copper microparticles. (**a**) Formation condition plot between applied voltage, *V*_crack_, and reference value of current, *V*_all_/*R*_res_. (**b**) Relationship between applied voltage and impedance after voltage application (0.10 kΩ). (**c**) Metal interconnect at 7.1 V_rms_. (**d**) Metal interconnect at 28 V_rms_. (**e**) Metal interconnect at 57 V_rms_.

**Figure 4 micromachines-14-00475-f004:**
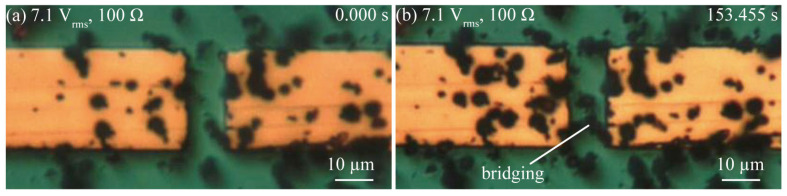
Series of optical images showing copper microparticle chain formation in the case of a low voltage (7.1 V_rms_, 100 Ω). (**a**) The beginning of the application of voltage to the gap. (**b**) Microparticles being trapped in the gap, thereby bridging the gap.

**Figure 5 micromachines-14-00475-f005:**
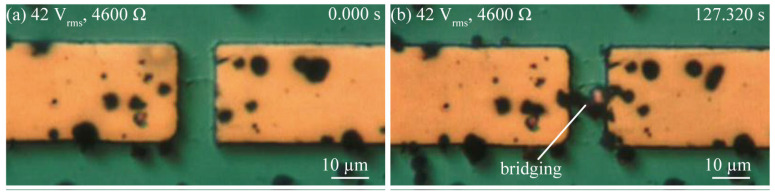
Series of optical images showing copper microparticle chain formation in the case of a high voltage with low reference value of current (42 V_rms_, 4600 Ω). (**a**) The beginning of the application of voltage to the gap. (**b**) Microparticles being trapped in the gap, thereby bridging the gap.

**Figure 6 micromachines-14-00475-f006:**
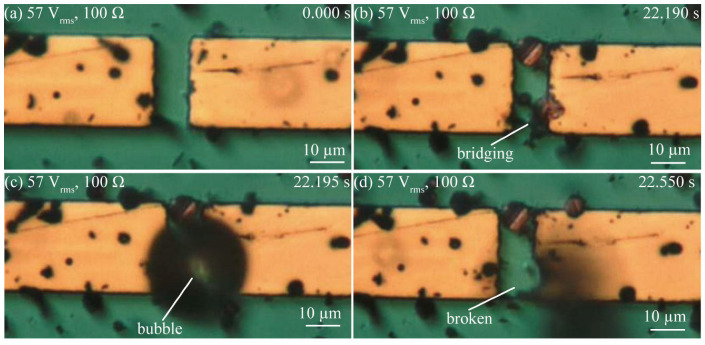
Series of optical images showing copper microparticles chain formation in the case of a high voltage with high reference value of current (57 V_rms_, 100 Ω). (**a**) The beginning of the application of voltage to the gap. (**b**) Microparticles being trapped in the gap, thus bridging the gap. (**c**) A bubble is generated when microparticles bridge the gap. (**d**) Microparticle chain and metal interconnect are broken.

**Table 1 micromachines-14-00475-t001:** Dimensions of the gold interconnect.

Gap Width [µm]	Interconnect Width [µm]	Interconnect Thickness [µm]
10	25	0.50

**Table 2 micromachines-14-00475-t002:** Material parameters of silicone oil.

	Conductivity [S/m]	Permittivity [F/m]	Viscosity [cSt]	Density [g/cm^3^]	Boiling Point [°C]
Silicone oil (KF-96-30cs, Shin-Etsu Chemical Co., Ltd., Tokyo, Japan)	1.00 × 10^−12^ or less	2.4 × 10^−13^	30	0.96	152

**Table 3 micromachines-14-00475-t003:** Material parameters of copper microparticles.

	Conductivity [S/m]	Permittivity [F/m]	Density [g/cm^3^]	Diameter [µm]
Copper microparticles (FMC-SB, Furukawa Chemical Co., Ltd., Osaka, Japan)	5.76 × 10^7^	7.7 × 10^−13^	3.5	3

## Data Availability

The data presented in this study are available in article.

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
