# Peer review of "Conduction Conditions for Self-Healing of Metal Interconnect Using Copper Microparticles Dispersed with Silicone Oil"

_micromachines, 2023, doi:10.3390/mi14020475_

Round 1
Reviewer 1 Report
The paper presents a self-healing electrical interconnect utilizing copper nanoparticle suspended in silicon oil and dielectrophoresis forces. While the concept is interesting, some modifications are necessary before the paper can be accepted. More quantitative results should be included. Details about the material and the experiment need to be included. Some of my other comments are listed below:
1. Please list all the dimensional parameters of the system (electrode size, nanoparticle diameter etc.) in a table. That will improve the readability of the paper.
2. Please lit all the material parameters of the system (conductivity, permittivity etc. of the oil, nanoparticle etc.).
3. Please discuss in detail why 32kHz frequency was selected. Apart from the abstract, the information is not presented anywhere else in the paper. As dielectrophoretic forces are a function of frequency, it should be discussed in the paper.
4. Please discuss the mechanism of the bubble formation.
5. Please include necessary information about the silicon oil (manufacturer, model number, cSt, density etc.) and the copper nanoparticle.
6. More quantitative results are needed to verify the claims of the paper. For example, a voltage sweep showing the transition between no conduction and conduction should be shown. That will show for each system type, what is the threshold voltage necessary for the nanoparticle bridging (and thus conduction condition) to form. Also, the measurements should be repeated for multiple frequencies.
7. Please calculate the Clausius-Mossotti factor for the nanoparticle silicon oil system.
8. A few more papers dielectrophoretic transport of micro/nanoparticles should be cited. I recommend the following:
a. https://doi.org/10.1021/ac070810u
b. https://doi.org/10.1021/acs.langmuir.2c02235
9. Please comment on the setup used for obtaining the experimental images (the microscope model, objective lens model, camera model etc.)
Reviewer 2 Report
The authors study dielectrophoretic trapping of copper microparticles as an approach to healing metal interconnects. They present a systematic study of the trapping process and identify three regions of operation depending on the applied voltage and induced Joule heating where particles can bridge a gap while not inducing conduction, particles bridging while allowing conduction, and induced instability due to high voltage leading to degradation of the particle-bridge and the interconnects. Publication of this manuscript is recommended upon addressing the following points and questions:
1. There are several typing and grammatical issues throughout the manuscript. Please address these prior to publication.
2. The caption to Figure 3 seem to be not in the correct order for the different sections.
3. Indicate the applied voltage on figures 4, 5 and 6 either on the figure itself or caption for clarity.
4. Please provide more details on the materials used. For example, it is helpful to specify the type of the silicone oil used. What is the oil’s thermal stability?
5. A more detailed discussion of the Joule heating is necessary. What temperatures do the authors expect are being reached to cause the disturbance of the silicone oil and degradation of the structures? A simulation of this process to provide more details to confirm the Joule heating hypothesis is helpful.
6. Upon bridge formation are the remaining suspension of oil and microparticles removed? If so, how is this accomplished?
Round 2
Reviewer 1 Report
The paper has been improved and can be considered for publication.